# Optimization of Z-shaped assembled mortise and tenon joint box culvert connection and structural characteristics

**Yuntao Yang**[ID]◐, **Hong Zhang***, **Longqi Liu**‡, **Baolei Zhu**‡, **Bingjie Song**‡

Transportation Institute, Inner Mongolia University, Hohhot, Inner Mongolia, China

◐ These authors contributed equally to this work.
‡ BZ and BS also contributed equally to this work.
* zhanghong3537@126.com

## Abstract

This paper proposes a new type of Z-shaped prefabricated mortise and tenon joint box culvert designed to enhance the longitudinal connection stability of highway prefabricated box culverts. Using actual engineering parameters and material properties, finite element simulation software was employed to analyze the culvert. The results indicated that maximum deformation and stress concentration occurred at the ends and armpits of the groove joint, specifically at the junction of the two culverts. The deformation of the middle section of the single culvert section is 75.86% higher than that of the end section. The deformation at the maximum deformation of the side wall of the single culvert section is 980% higher than that at the end. Under identical soil filling conditions, the mechanical properties of the integral box culvert and the prefabricated box culvert exhibited the greatest differences at their joint. The deformation and stress values at other locations showed minimal differences. Based on the stress and deformation characteristics of the culvert body, various box culvert designs with different joint heights and lengths were developed and simulated. The findings indicate that the mechanical properties of the joint culvert are most significantly enhanced with a joint height of 1.8 m and a joint length of 0.5 m. By integrating the advantages of the mortise and tenon joint structure—such as limiting joint displacement, preventing misalignment and uneven settlement, and facilitating smooth load transfer—the mortise and tenon joint is incorporated into the assembled box culvert joint. Ultimately, a Z-shaped prefabricated mortise and tenon joint box culvert, featuring a well-designed structure and a 1.25 m long mortise and tenon joint, was selected. Following optimization, the mechanical properties of the assembled box culvert joints have significantly improved. The connection performance is excellent, meeting the required bearing capacity, and demonstrating strong compressive shear and deformation resistance. Additionally, the yield and failure limits under identical uneven loads have been considerably enhanced, while maintaining economic and

**Data availability statement:** All relevant data are within the article and its Supporting information files.

**Funding:** The author(s) received no specific funding for this work.

**Competing interests:** The authors have declared that no competing interests exist.

prefabrication rationality. This study provides valuable insights for the practical engineering of similar assembled box culverts.

---

## 1. Introduction

In highway engineering construction, the prefabricated box culvert, a reinforced concrete structure designed for multiple functions such as drainage and passage, has become the primary construction method for culvert engineering. This shift is attributed to its advantages, including efficient construction, time savings, improved component quality control, and significant economic benefits.

In the exploration of traditional underpass mechanisms, particularly box culverts, researchers have accumulated substantial experience. Acharya et al. [1] conducted static and traffic load tests on reinforced concrete box culverts across three sections(concrete pavement, concrete shoulder, and unsurfaced fill) to investigate the influence of factors such as load size, mode, speed, and road structure on culvert response. The results indicate that under static load, the deflection of the culvert roof increases sequentially from concrete pavement and concrete shoulder to the faceless fill layer. A truck traveling at high speed typically causes greater deflection than a truck moving at low speed or stationary. Singh et al. [2] conducted a design and analysis of a two-compartment reinforced cement concrete (RCC) culvert and found that as the size of the culvert decreased, its weight significantly reduced. Additionally, the articles in the IRC specifications were thoroughly discussed and considered. Lee et al. [3] analyzed the horizontal earth pressure on underground box culverts and conducted centrifuge model tests for accurate simulation. They concluded that the horizontal earth pressure acting on the sides of the box culvert exhibits significant variability when only the earth pressure coefficient and the soil friction angle are considered. Therefore, a correction coefficient is necessary for calculating the horizontal earth pressure on the sides of the box culvert. Abuhajar et al. [4] utilized RPI geocentrifuge equipment and a one-dimensional seismic simulator to investigate the seismic response of uniform sandy soil at 60 g, as well as the seismic response of two soils with different relative densities (Dr = 50% and 90%). They also examined the seismic soil-structure interaction involving sandy soil, a hidden box culvert, and a surface foundation. Huh et al. [5] developed a simplified yet comprehensive method for assessing the vulnerability of shallow, two-story reinforced concrete underground box structures built in highly weathered soil, represented through vulnerability curves. Additionally, the results of each peak ground acceleration (PGA) earthquake were compared to identify the most valid and appropriate number of cost-benefit analyses in terms of time and resources. McGuigan et al. [6] utilized centrifugal test results to validate numerical models that assess culvert spacing and the geometry of compressible zones for double-forward protrusions and induced gully culverts. To address the significant vertical pressure, Sun et al. [7] constructed a 2-foot-thick trench above a rigid concrete box culvert, which was reinforced at a highway site in Kentucky. They conducted on-site measurements of stress, strain, and geofoam settlement over a

period of five years. Wang et al. [8] employed the box-culvert shield roof method to construct an underpass for a new road beneath an existing road in Zhengzhou, China. They also investigated the settlement characteristics of the existing road. A three-dimensional numerical model was developed using numerical analysis methods and validated through field measurements. Flener et al. [9] conducted a static load analysis on four large-span deep corrugated steel box culverts, with spans of 14 m and 8 m. Their study confirmed that the plain structure is more sensitive to the effects of covering depth compared to the reinforced structure. Shatnawi et al. [10] conducted a numerical simulation to investigate the mechanical properties of reinforced concrete box culverts with various structural designs under soil filling loads. Maximos et al. [11] investigated the mechanical performance and fatigue behavior of reinforced concrete box culverts subjected to cyclic loading through model experiments, finding that the flexural capacity of these culverts remained largely unaffected by fatigue.

Among the various research topics related to box culverts, scholars have primarily focused on the interaction between box culverts and soil. Abuhajar et al. [12] conducted small centrifuge physical model tests to investigate the factors influencing the soil arch effect in box culverts. They found that these factors are related not only to the height of the soil column above the culvert but also to the culvert's thickness, the soil's modulus of elasticity, and the Poisson's ratio. Wood et al. [13] analyzed two production-oriented models for culvert load grade demand using live load test data from a measured box culvert with four different cover soil depths. Compared to the structure-frame model, the complexity of the soil-structure model is increased, resulting in improved accuracy of the predicted torque. Improvements in modeling predictions related to increased complexity occur only when structure-frame models are highly imprecise. To reduce the load exerted by the subgrade soil on the box culvert structure, Chen et al. [14] proposed a novel load-shedding culvert design and investigated its soil pressure distribution through model tests. Building on this foundation, the stress states of three culvert structures and their significant influencing factors were analyzed using numerical simulations. The study reveals that the LSC, compared with EBC and IBC, can not only reduce the vertical earth pressure on the top slab but also reduce the horizontal earth pressure on the culvert sidewall. (C) 2016 Elsevier Ltd. All rights reserved. Debiasi et al. [15] concluded that shallow-buried rectangular structures are significantly influenced by nonlinear friction effects at the soil-structure interface. Additionally, the soil-structure interaction transitions steadily under earthquake conditions from a deeply buried state to a "zero coverage depth" state. Oshati et al. [16] presented field-measured earth pressure data for a rectangular box culvert constructed with cast-in-place steel bars. Their findings indicated that the downward resistance along the side walls of the culvert contributed to an increase in foundation pressure. Sun et al. [17] conducted on-site blasting and vibration tests on the blasting project of the connecting passage of Wuhan Metro Line 8, and used LS-DYNA software to analyze the dynamic response characteristics of an underground drainage box culvert during the blasting test. The evolution law of vibration response of buried drainage box culvert under blasting vibration is studied, and the safety surface control standard of blasting vibration of drainage box culvert is put forward.

Prefabricated box culverts, as a novel construction method, offer several advantages, including a short construction period, ease of quality control, and material savings. These benefits have led to their increasing adoption in underground engineering projects, making them a focal point of scientific research. Gong et al. [18] conducted experimental research on two types of prefabricated box culverts. They analyzed the damage process of the box culverts using the traditional fuzzy C-means method (FCM), based on the acoustic emission parameters. Additionally, they proposed an improved algorithm, G-DFCM, which combines mesh density and distance. Xu et al. [19] performed a finite element analysis to evaluate the bearing capacity of a prefabricated PBC specimen designed for box culverts subjected to four-point bending. The study also discusses the bearing capacities of three additional types of concrete, which share the same reinforcement ratio but differ in reinforcement configurations. De Nóbrega et al. [20] investigated the dynamic performance of prefabricated box culvert underpasses designed for high-speed railways. A two-dimensional finite element model was developed using the ANSYS program, based on the road-structure transition proposed by ADIF.

The prefabricated box culvert consists of sections that are assembled longitudinally. While it is more efficient and cost-effective than traditional cast-in-place box culverts, it has several drawbacks, including numerous joints, reduced

overall stiffness, and the potential for collisions and damage during transportation and construction. Consequently, the structural form of the sections, the mode of longitudinal connections, and the strength of these connections significantly influence the overall stability of the culvert. The road performance and service life of the culvert are critical factors that determine its effectiveness.Currently, most assembled culverts, both domestically and internationally, utilize flat section joints. These joints are secured with external structures such as bent bolts, prestressed steel strands, and steel rods to ensure stable longitudinal connections. Additionally, waterproof designs are implemented at the joint connections. However, there remains a lack of systematic research on methods to enhance the stability of longitudinal connections in prefabricated box culverts through the use of specialized structural forms and connection modes.

This paper presents a new splice mode for fabricated box culverts, drawing on current research and previous experience. It incorporates the "mortice and tenon structure," a technique traditionally used by ancient Chinese craftsmen, into the joint design. The finite element numerical simulation method was employed to enhance and optimize the new structural form and connection mode, resulting in a culvert structure that better supports the longitudinal connections of fabricated box culverts. This research offers valuable model references and data support for the design of box culverts in highway engineering.

## 2. Z-shaped assembled box culvert with a mortise and tenon joint structural design

This article presents a Z-shaped prefabricated box culvert designed with a reinforced concrete structure for highway applications. The figure illustrates the three-dimensional schematic diagram, as well as the front and side views of a single culvert section (units: mm). The overall shape of the box culvert resembles the letter "Z." The culvert body has a net height of 3.0 m, with top and bottom plates each measuring 3.0 m in width. The total length of a single culvert section is 5.0 m. The thickness of the top plate, bottom plate, and side walls is 0.3 m, while the opening has a square cross-section with a side length of 2.4 m. The design of the four edges surrounding the culvert body features a 135° cut at the armpit angle. This design aims to prevent excessive stress concentration and minimize collision damage during construction (Fig 1).

This culvert is classified as an integral assembled box culvert based on its basic prefabricated form. The connections of the culvert body utilize a staggered splicing design. The front end of the culvert section extends 1.0 m in length and features a suspended structure that is half the height of the culvert. The rear end extends an equally long lower structure to ensure that the tail of the culvert body provides firm support for the subsequent culvert section. The joint features a semi-circular groove with a diameter equal to the thickness of the side wall, comprising a raised upper section and a concave lower end. The Z-shaped staggered splicing mode provides overlapping support at both the front and back, thereby reducing settlement and displacement of the culvert structure. Conversely, it can increase the contact area among different culvert joints, enhance interface friction, and significantly improve the overall stability of the longitudinal connections in the culvert. Additionally, the circular arc groove joint can further enhance the interface connection and effectively restrict the lateral displacement of the culvert body, thereby providing effective restraint. The schematic diagram illustrating the connection mode between culvert sections is presented in the figure, using two sections as an example (Fig 2).

## 3. Numerical analysis of the mechanical properties of prefabricated splicing culverts

### 3.1. Numerical simulation of fundamental mechanical properties

#### (1) Establishment of the model and application of loading

This paper aims to thoroughly analyze the fundamental mechanical properties of a new type of prefabricated splicing culvert, verify the stability and reliability of the structure, and provide data support for optimizing the structural design and integrating innovative features. To achieve this, Ansys Workbench finite element numerical simulation software is utilized to establish a 1:1 three-dimensional model of the box culvert and a solid model of the culvert soil using actual engineering material parameters, as illustrated in the (Figs 3 and 4).

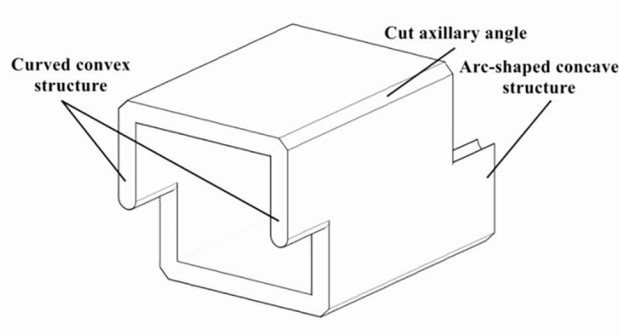

(a)Three-dimensional structural diagram

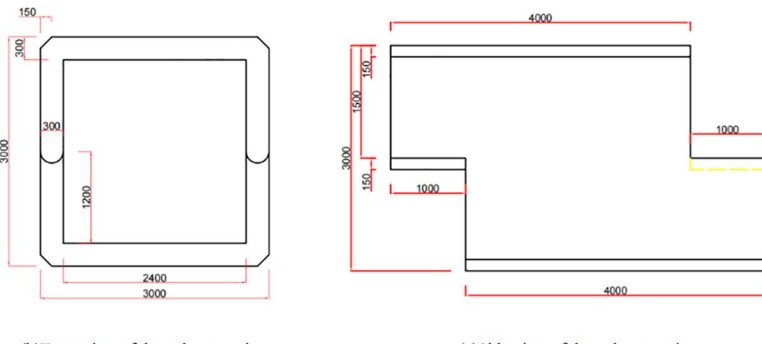

(b)Front view of the culvert section

(c)Side view of the culvert section

**Fig 1. Three-dimensional structure and cross-sectional dimension diagram of a Z-shaped culvert.**

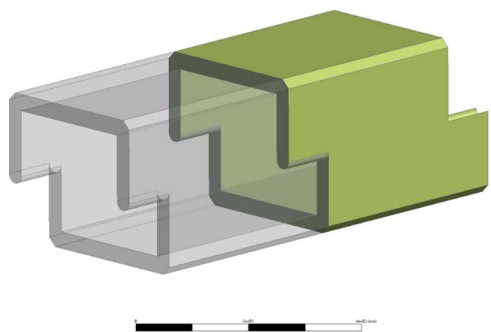

**Fig 2. Schematic diagram illustrating the connection effects of a Z-shaped culvert.**

The specific parameters of the model materials are presented in the following Table 1:

The modeling parameters are presented in Table 2.

In the ANSYS finite element software, the grid division method of the model is structured grid, and the method is sweeping. This kind of grid unit arrangement rules, has a clear topological structure. The relationship between nodes and units is defined by index (i, j, k), which has high computational efficiency and is convenient for storing and accessing data.

(Due to the corresponding subject of this study is prefabricated box culvert for highway, the material parameters applied in the model, including enhanced elastic modulus, Poisson 's ratio and other data, are derived from Chinese domestic

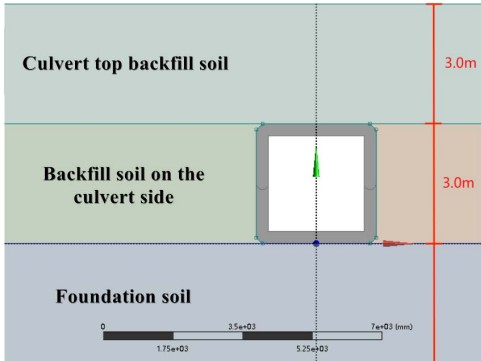

**Fig 3. Diagram of the structural model.**

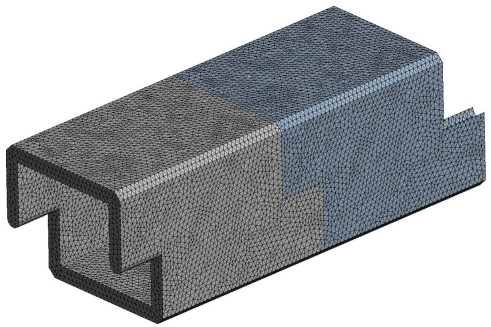

**Fig 4. Diagram of grid division for the double-segment culvert body.**

**Table 1. Table of Material Parameters.**

| Concrete | | Rebar | | Foundation soil and backfill soil | | |
|---|---|---|---|---|---|---|
| Parameter | Value | Parameter | Value | Parameter | Value | |
| Strength grade | C40 | Main reinforcement diameter/mm | 20 | Elastic modulus/MPa | 50 | 25 |
| Elastic modulus/MPa | $3.25 \times 10^4$ | Stirrup diameter/mm | 10 | Poisson's ratio | 0.4 | 0.4 |
| Poisson's ratio | 0.2 | Elastic modulus | $2 \times 10^5$ | severe/kg/m$^3$ | 18 | 18 |
| | | | | friction angle/° | 25 | 30 |
| severe/kg/m$^3$ | 25 | Poisson's ratio | 0.2 | Cohesive force/KPa | 25 | 50 |

codes such as *JTGT 3365-02-2020 Highway Culvert Design Code and Technical Specification for Design* and *Construction of Full-face Prefabricated Reinforced Concrete Box Culverts (DB41 _ T 2279-2022)*. The data values meet the previous research experience.)

To simulate the actual construction conditions of prefabricated box culverts, the original single culvert section is utilized as the loading body, with a backfill height of 6.0 m and a top cover soil height of 3.0 m. The fill soil used in actual construction does not exceed this height. (In the process of modeling, in order to deeply reflect the connection mode between culvert joints, maximize the reduction of real materials, and make the simulation results more scientific, the following assumptions are adopted for the finite element model: considering the nonlinear behavior of materials (such as plasticity,

**Table 2. Table of Modeling Parameters.**

| Modeling project | Modeling details | |
|---|---|---|
| **Model Category** | **Discrete model** | |
| Contact settings | Reinforcement – Concrete | penalty function |
| | Concrete components | Face to face contact |
| | | Concrete – backfill soil |
| Culvert unit setting | Unit Type | concreteSOLID65 |
| | | rebarPIPE8 |
| | Number of nodes | 123064 |
| | Number of units | 43268 |

hyperelasticity, viscoelasticity, etc.). Large deformation assumption: considering geometric nonlinear effects (such as large displacement, large rotation). Fixed boundary conditions: It is assumed that the displacement of the fixed end is completely constrained.)

The Newton-Raphson iterative method (fully N-R processing method) is employed to solve the specified settings. Establish multiple monitoring points at each section to assess the deformation of the culvert body and the distribution of stress and strain under soil pressure. The calculation results are obtained by inserting total deformation probes and equivalent stress probes at the monitoring points. The distribution of monitoring points and probes is illustrated in Fig 5. The Newton-Raphson iterative method (fully N-R processing method) is employed to solve the specified settings.

**(2) Analysis and simulation results of body deformation and stress**

Fig 6 illustrates the simulation results of the mechanical properties of the culvert body in a filled soil state.

When subjected to top and lateral earth pressure, along with the constraints of the front and rear culvert joints, the maximum deformation of the roof at a filling height of 6.0 m is $1.8833 \times 10^{-2}$ mm. This maximum deformation occurs at the center of the cross-section and gradually increases from the mid-span section to the culvert end section. The roof experiences the gravitational force of the overlying soil, while its two ends are connected to the tops of the side walls, which provide an upward support reaction force. In contrast, the central position remains unsupported. Under the influence of self-weight and earth pressure, the upper side of the plate experiences compression, while the lower side undergoes tension, resulting in significant downward deformation at the central position.

The maximum deformation and stress concentration in the side wall are prominently observed at the ends and armpits of the joint, specifically at the connection between the two culvert joints. The end resembles an extended cantilever beam structure, which is directly in contact with and compressed by the front and rear culverts. The deformation and stress in this area are more significant than in other parts, making it susceptible to impact damage during transportation and construction. The angle at the axillary position is 90 degrees, which experiences significant shear force under load, resulting in noticeable stress concentration. This area is the weakest part of the entire culvert. In actual projects and during long-term service, this area is more susceptible to cracking, water leakage, and other issues compared to other parts.

**3.2. Comparative analysis of spliced culverts and whole culverts**

To comprehensively compare the traditional integral cast-in-place culvert with the assembled splicing culvert, highlight the superiority of its longitudinal connection capability, and validate the practical value of the assembled splicing culvert project, an integral culvert model is established. This model has the same height and double the section length, based on the dimensions of the Z-shaped splicing culvert and the material parameters of the model. This approach effectively connects the two-segment assembled box culverts into a single unit. To ensure the reliability of the comparison data, both ends of the integral culvert are designed with the same joint structure as the assembled culvert. The integral culvert is then

**Fig 5. Distribution map of monitoring points and probes.**

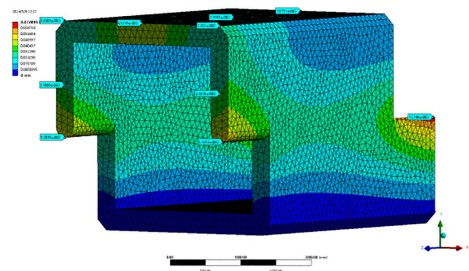

（a）The total deformation cloud map of a single culvert section in the filling state

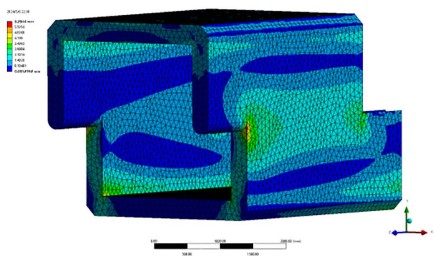

（b）Equivalent stress cloud diagram of a single culvert section in the filling state

**Fig 6. Simulation results of the mechanical properties.**

subjected to the same load as the assembled box culvert for simulation calculations. The figure illustrates the simulation results of the mechanical properties (Fig 7).

The deformation and stress cloud diagram indicates that, under the same filling conditions, the greatest difference in mechanical properties between the integral box culvert and the assembled box culvert occurs at the joint of the prefabricated culvert. In contrast, the deformation and stress values at other locations show minimal variation. The deformation and stress at the joint of the Z-shaped prefabricated culvert are significantly greater than those in the middle section of the integral culvert. This indicates that the prefabricated construction method offers advantages such as a shorter construction period and lower costs compared to the traditional cast-in-place integral culvert. Simultaneously, this approach increases the weak areas, compromises the integrity of the culvert, and diminishes the mechanical properties of the box culvert. The design optimization of the prefabricated box culvert should prioritize enhancing the stability of the interface connection and

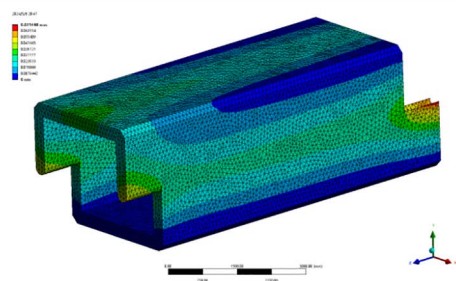

(a) Total deformation cloud diagram of the integral culvert section

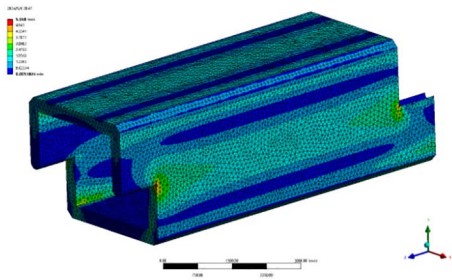

(b) Equivalent stress cloud diagram of the integral culvert

**Fig 7. Simulation results of mechanical properties of integral culvert.**

minimizing stress concentration in the weak areas. The figure presents a comparison of the mechanical performance data between the integral box culvert and the assembled lap culvert (Fig 8).

The data comparison diagram indicates that both the assembled box culvert and the integral box culvert exhibit small deformation at the ends and significant deformation in the middle, with no notable differences in deformation and stress values. The deformation value of the assembled lap culvert at the joint position, located in the middle of the side wall, is 224.62% higher than that of the integral box culvert. Additionally, the equivalent stress value is 93.51% higher than that of the integral box culvert. However, since the deformation value is minimal, the equivalent stress is significantly lower than the ultimate strength of the culvert material. Therefore, a comparison is made between the assembled lap culvert and the integral culvert. This disadvantage is deemed acceptable and still holds considerable practical value.

## 4. Optimization of the Z-shaped lap culvert structural form

Based on the deformation and stress simulation results of the new culvert design, the optimization of the structural form of the assembled Z-shaped lap culvert is conducted. This optimization aims to reduce the deformation of each component, thereby enhancing its anti-deformation capacity during construction and service. Additionally, it seeks to lower the stress values at points of stress concentration to improve its resistance to damage, which includes optimizing the height and length of the connecting ends.

### 4.1. Optimization of the Z-joint height

To address the significant central deformation and minimal deformation at both ends of the side wall joint of the assembled Z-shaped lap culvert, the design optimizes the side wall force by increasing the joint height, taking into account the

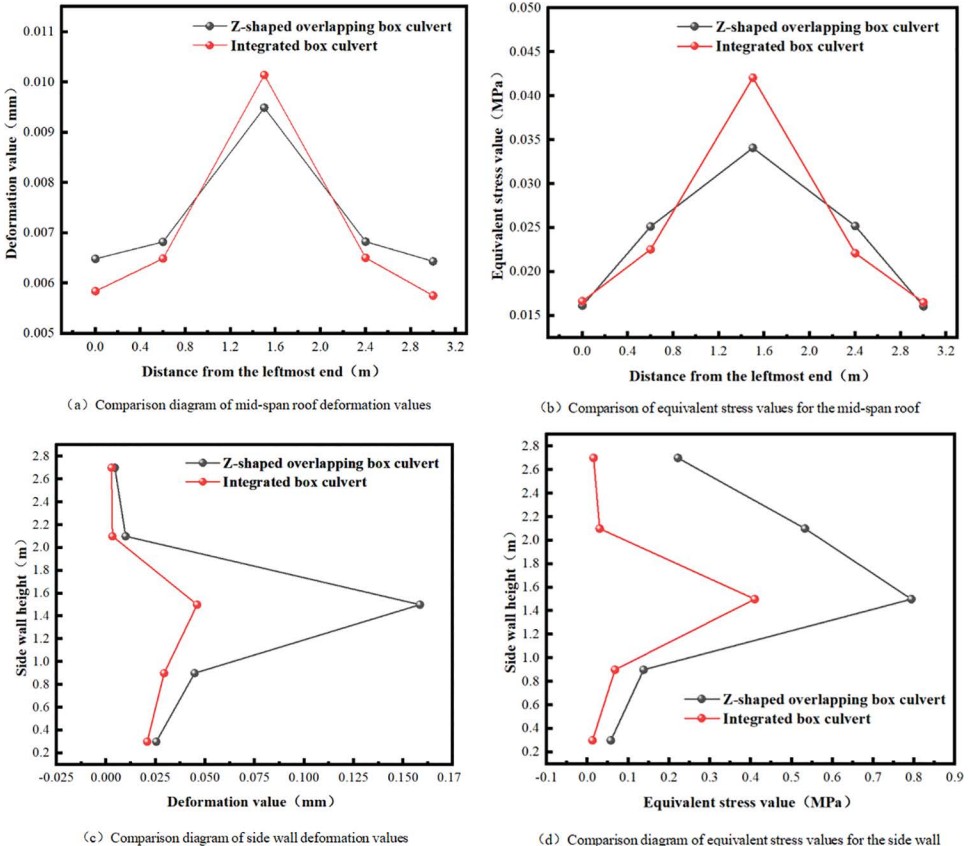

（a）Comparison diagram of mid-span roof deformation values

（b）Comparison of equivalent stress values for the mid-span roof

（c）Comparison diagram of side wall deformation values

（d）Comparison diagram of equivalent stress values for the side wall

**Fig 8. Comparison of mechanical properties data.**

characteristic of soil pressure increasing with depth. Three different joint heights have been designed, as illustrated in Fig 13. The vertical heights of the head convex end from the ground are 1.5 m, 1.8 m, and 2.1 m (Fig 9).

Ansys Workbench was utilized to establish the model, and the soil was simultaneously filled to a depth of 6.0 m for loading analysis. Stress and strain data were extracted, and the deformation and stress values at the central joint of the side wall were monitored (Fig 10).

Under the condition of fully covered soil, as soil pressure increases with depth, shallower depths result in reduced lateral extrusion of the soil on both sides. The deformation and equivalent stress at the side wall joint of the Z-shaped prefabricated box culvert decrease as the joint height increases. When the height is approximately 1.8 m and the net height of the upper convex joint is 1.2 m, the deformation and equivalent stress attain their minimum values. The deformation value is 36.197% lower than that of the original design, while the stress value is 26.411% lower. When the height is further increased, the net height of the upper end decreases. This is analogous to a cantilever beam subjected to a large bending moment, which results in increased deformation and equivalent stress due to its reduced strength.

Therefore, considering the challenges of prefabrication, material costs, and construction convenience, a joint form with a height of approximately 1.8 m from the ground should be adopted to optimize the stress distribution of the culvert. This approach will enhance the strength and deformation resistance of the box culvert's side wall and prevent damage during assembly and service.

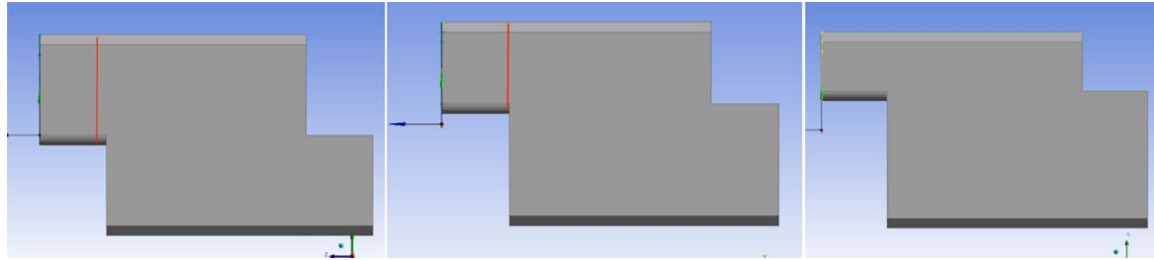

**Fig 9. Comparison of Models with Different Joint Heights.**

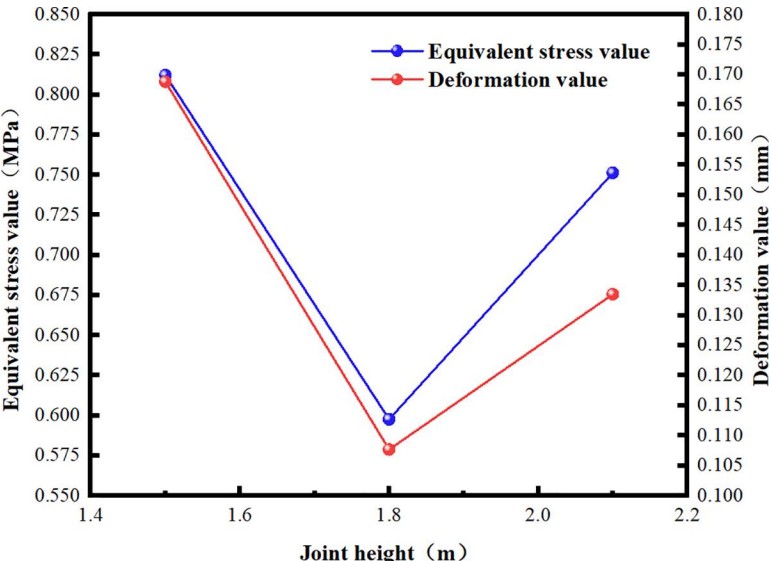

**Fig 10. Deformation and Stress Values at Joints with Varying Heights.**

## 4.2. Optimization of the Z-joint length

Considering that the maximum deformation and stress concentration in the culvert occur at the ends and armpits of the joint—specifically, the connection between the two culvert joints—the length of the original joint is shortened without affecting the longitudinal connection. This modification reduces the length of the structural beam, similar to a cantilever beam, improves shear resistance, mitigates the adverse effects of the self-weight at the connection end, and enhances structural stability. Based on the selected joint height of 1.8 m in Section 4.1, the original joint is shortened transversely to reduce the length of the single culvert section. The four joint lengths of 1.0 m, 0.75 m, 0.5 m, and 0.25 m at the connection end (Fig 11) are compared to the original culvert.

The model was established using Ansys Workbench, and the soil was simultaneously filled to a depth of 6.0 m for loading analysis. Stress and strain data were extracted, along with the deformation and stress values of the key joint components (Fig 12).

Under the same conditions of fully covered soil, shortening the joint length results in a decrease in joint weight and cantilever length, thereby improving the mechanical properties of the joint. When the joint length is 0.5 m, the deformation and equivalent stress values reach their minimum and subsequently begin to increase. When the joint length is too short,

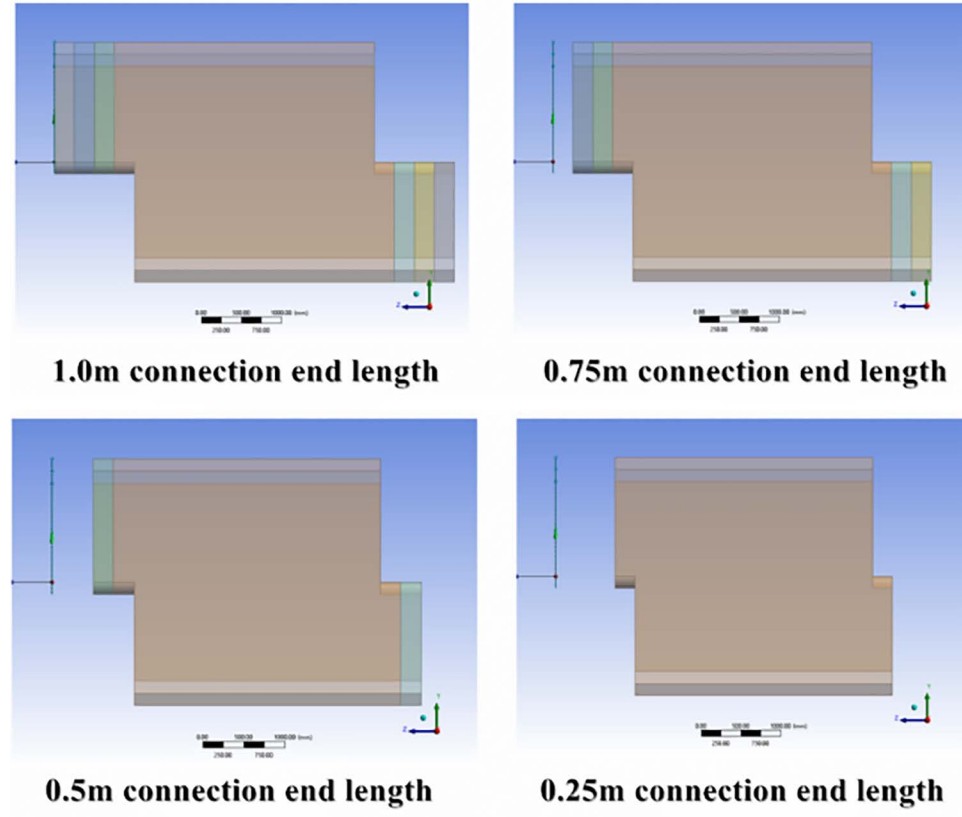

**Fig 11. Schematic Diagram of Connector Reduction.**

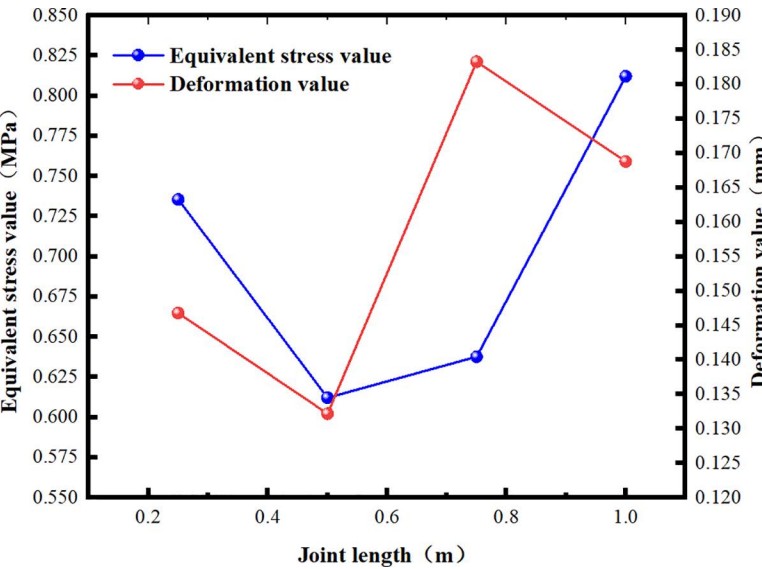

**Fig 12. Deformation and Stress Values at Joints with Varying Lengths.**

interface stress concentration becomes significant, preventing the longitudinal connection stability advantages of the Z-shaped lap form from being realized, which makes the structure prone to dislocation. Based on the simulation results and actual construction requirements, a joint length of 0.5 m is preferred as the final structural form.

## 5. Optimization of the longitudinal connection mode of the Z-shaped culvert mortise and tenon structure

### 5.1. Overview of the design of box culvert mortise and tenon structures

The mortise and tenon joint structure is an ancient Chinese construction technique that primarily uses wood, brick, and tile as building materials, with a wooden frame as the main structural method. The structure consists of columns, beams, purlins, and other primary components. The joints between the various components utilize a mortise and tenon design, creating an elastic frame. This highly innovative invention transforms the traditional Chinese wooden structure into a unique flexible system that surpasses contemporary architectural bents, frames, and steel structures. It can withstand large loads while allowing for some deformation. Under seismic loading, a portion of the seismic energy is absorbed through deformation, thereby reducing the structure's seismic response.

This paper integrates the characteristics of the Z-shaped lap culvert structure with a mortise and tenon design in the prefabricated box culvert joint. The structure is illustrated in the accompanying diagram, where a cylindrical 'convex tenon' is positioned at the protruding end, and a corresponding 'concave mortise' of the same size is located at the concave joint end of the subsequent culvert section. High-strength steel bars are embedded within the mortise and tenon structure of the cylinder to improve its shear strength. Joint reinforcement at the socket is achieved using a water expansion water stop belt, a water stop rubber ring, and other mortar bonding materials (Fig 13).

During the mutual insertion of the mortise and tenon structure at the culvert joints, the convex tenon structure functions similarly to a dowel bar. This design not only restricts the horizontal and vertical displacement of the culvert joints and prevents dislocation and uneven settlement of the culvert body, but also facilitates smooth load transmission, enhances shear strength at the joints, and significantly improves the mechanical properties and practicality of the culvert body. The cross-sectional arrangement of the socket column (hole) is illustrated in the figure (Fig 14).

To thoroughly investigate the actual effects and mechanical properties of the mortise-tenon socket structure used in the Z-shaped splicing culvert, and to enhance the performance of this structure, the length of the convex mortise-tenon socket column was selected as a variable for simulation optimization.

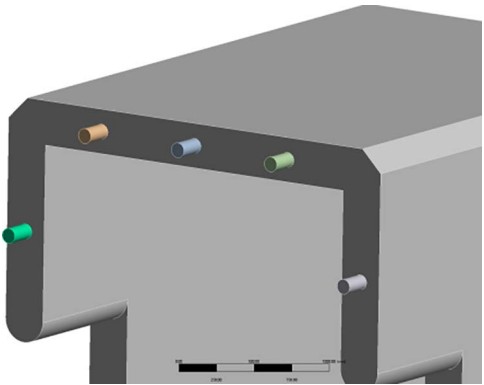

**Fig 13. Schematic diagram of the mortise and tenon joint structure at the culvert end joint.**

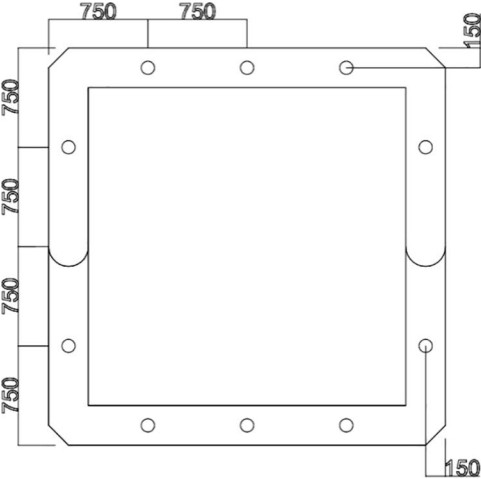

**Fig 14. Cross-sectional arrangement of the socket column (hole).**

### 5.2. Optimization of the length of the convex tenon socket column

The short tenon socket column fails to fully utilize the advantages of horizontal and vertical fixation provided by the mortise and tenon structure. Additionally, it complicates the incorporation of other waterproof materials and is susceptible to collision damage during transportation and construction, which may lead to overall failure. Therefore, using a minimum length of 50 mm, seven variables were established for simulation: 50 mm, 75 mm, 100 mm, 125 mm, 150 mm, 175 mm, and 200 mm. The roof of the culvert, along with the root, middle, and end of the socket column, were selected as observation points for deformation and stress. The variations in mechanical properties under external soil load were then investigated and optimized (Fig 15).

As the length of the socket column (hole) gradually increases from 0 mm, the deformation of the culvert roof exhibits a slight upward trend. At a length of 125 mm, the roof deformation value reaches its minimum. This phenomenon arises from the presence of the socket column, which restricts the vertical displacement of the roofs of the two culverts and serves a stabilizing function. As the length of the socket column increases, the restriction and fixation become more significant, resulting in reduced deflection under the same load. When the length of the socket column is 125 mm, the deformation of the middle section of the roof (1.5 m from the leftmost end) is 24.131% lower than that at 0 mm. Beyond 125 mm, the roof deformation stabilizes and exhibits a slight increase as the length of the socket column increases. This phenomenon occurs because the socket column resembles a cantilever beam structure with one end fixed. As its length increases, both the shear force and bending moment increase simultaneously, leading to fluctuations in roof deformation (Figs 16–18).

Analyzing the deformation at the root, middle, and end positions of the socket column in relation to the socket length reveals that the convex tenon socket column with a length of 125 mm best meets the design and construction requirements.

### 5.3. A comparison of force characteristics under identical loading conditions

To further investigate the beneficial effects of the mortise and tenon joint structure on the overall strength and connection performance of prefabricated components, particularly regarding joint stability, numerical simulation software was employed for additional simulations and verification.

In Ansys Workbench finite element numerical simulation software, two sets of 1:1 three-dimensional box and culvert joint models are created based on actual engineering material parameters, with a total of two culverts. One group

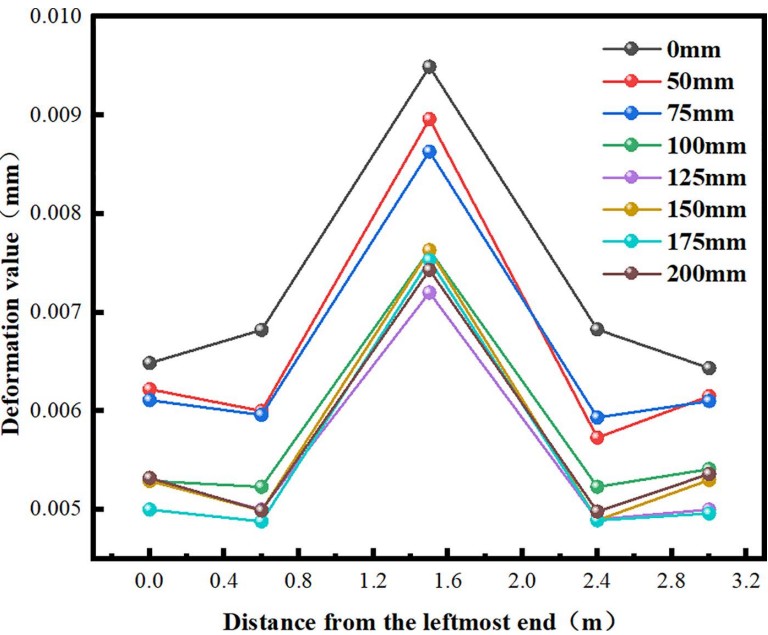

**Fig 15. Diagram of Roof Deformation.**

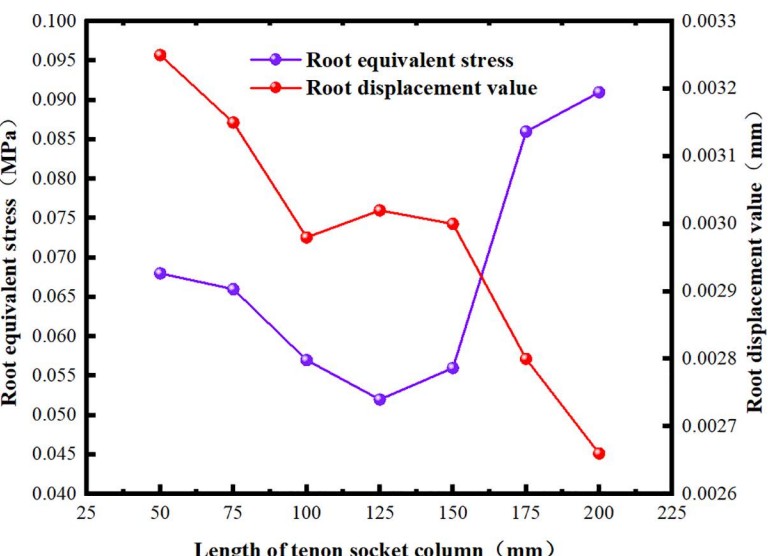

**Fig 16. Equivalent Stress and Deformation Values at the Root of the Socket Column.**

features a mortise and tenon socket structure (as illustrated in the figure), while the other group lacks this structure. Aside from the structural form, the geometric dimensions, material parameters, and boundary conditions of both model groups are consistent. The loading tests are conducted in accordance with actual engineering practices for box culvert service (Figs 19 and 20).

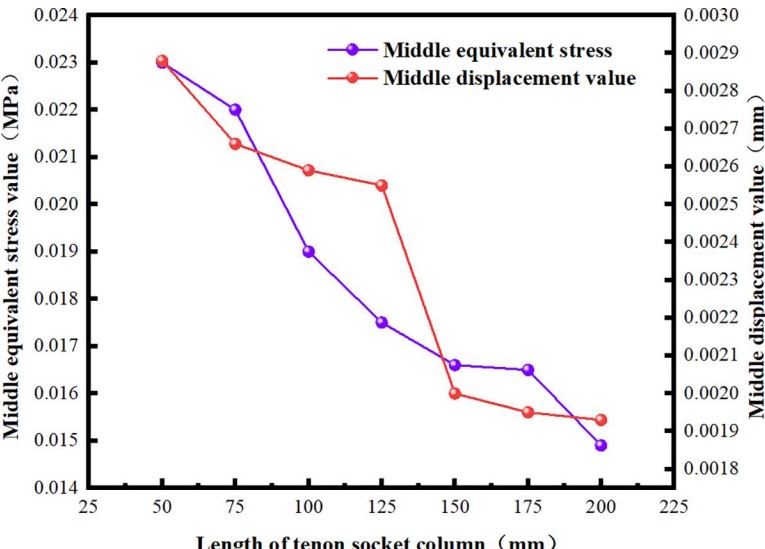

**Fig 17. Equivalent Stress and Deformation Values at the Midpoint of the Socket Column.**

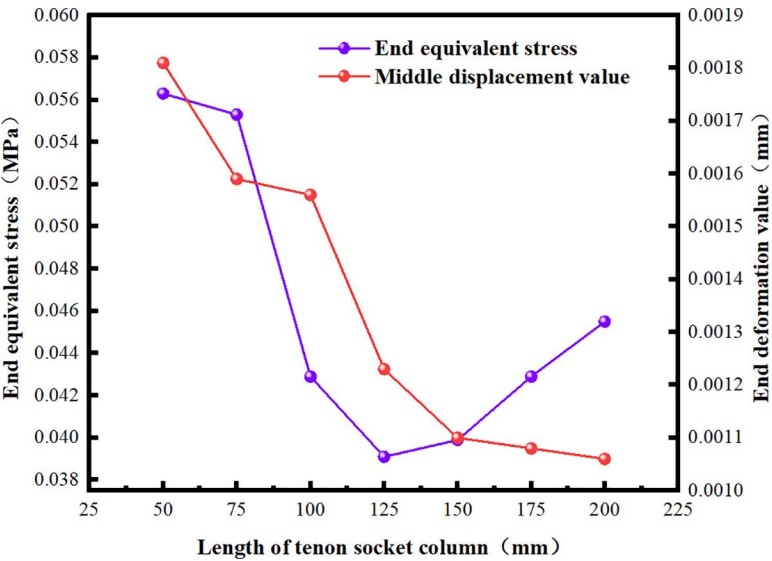

**Fig 18. Equivalent Stress and Deformation Values at the End of the Socket Column.**

(The diagram of the joint model lacking a mortise and tenon structure will not be repeated due to the previously mentioned morphology.)

According to the General Specifications for the Design of Highway Bridges and Culverts (JTG D60-2015), the functions utilized in the design of highway bridges and culverts are categorized into four types: permanent, variable, accidental, and seismic.Different representative values must be assigned to each category based on the following provisions:

1. The representative value of a permanent action is defined as its standard value.The standard value of a permanent action can be determined through statistical analysis, calculations, and a comprehensive evaluation that incorporates engineering experience.

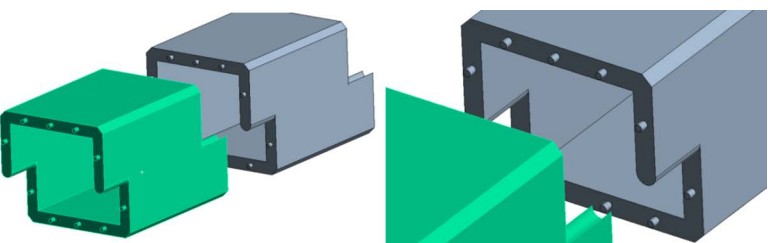

**Fig 19. Overall model diagram of two culverts featuring a socket device in a separated state.**

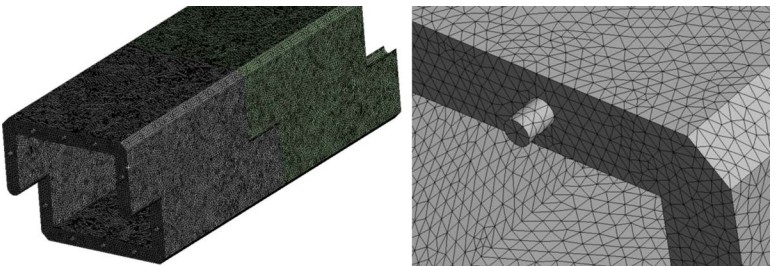

**Fig 20. The inclusion grid separates the overall structure from its detailed components.**

2. The representative values of variable actions include the standard value, combination value, frequency value, and quasi-permanent value.The combination value, frequency value, and quasi-permanent value can be calculated by multiplying the standard value of the variable action by their respective coefficients: combination value coefficient ($\psi_c$), frequency value coefficient ($\psi_f$), and quasi-permanent value coefficient ($\psi_0$).

3. The design value of accidental action is considered the representative value, which can be determined based on historical records, field observations, and tests, along with a comprehensive analysis of engineering experience. Additionally, it may be established according to the specific provisions of relevant standards.

When designing the highway bridge and culvert structure according to the limit state of bearing capacity, the basic combination of actions should be used for both permanent and temporary design conditions. The accidental combination of actions should be applied for accidental design conditions, while the seismic combination should be utilized for seismic design conditions. These combinations must comply with the following provisions:

The fundamental combination consists of the design value for permanent actions and the design value for variable actions.

$$S_{ud} = \gamma_0 S \left( \sum_{i=1}^{m} \gamma_{G_i} G_{ik}, \gamma_{Q_1} \gamma_L Q_{1k}, \psi_c \sum_{j=2}^{n} \gamma_{Lj} \gamma_{Qj} Q_{jk} \right)$$

$$S_{ud} = \gamma_0 S \left( \sum_{i=1}^{m} G_{id}, Q_{1d}, \sum_{j=2}^{n} Q_{jd} \right)$$

When the action and its effects are considered to be linearly related, the design value of the combined action effects can be calculated by summing the algebraic values of the action effects.

All effects calculated based on the limit state of bearing capacity design specified in the code are converted to a uniform load and applied to the two groups of combined inclusion bodies. The resulting stress-strain and deformation values are presented in the figure (Figs 21 and 22).

The calculation results were observed and extracted by placing a total deformation probe and an equivalent stress probe at the monitoring point. Specific locations are illustrated in the figure below (Figs 23–25).

A. The central section connecting the end of the body.

B. Located above the left socket column of the culvert body.

C. Located above the right socket column of the culvert body.

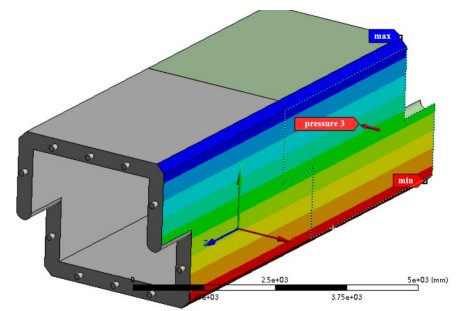

**Fig 21. Lateral loading mode (refers to the method of applying forces parallel to the ground).**

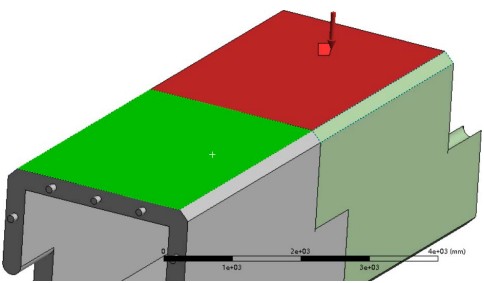

**Fig 22. The model for converted uniform pressure loading.**

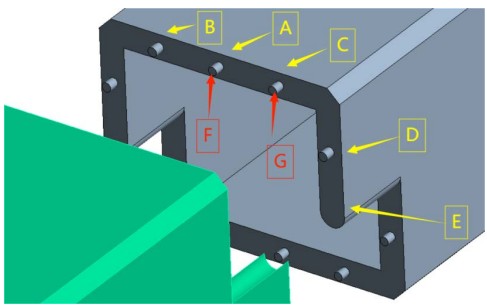

**Fig 23. Map indicating the locations of measuring points.**

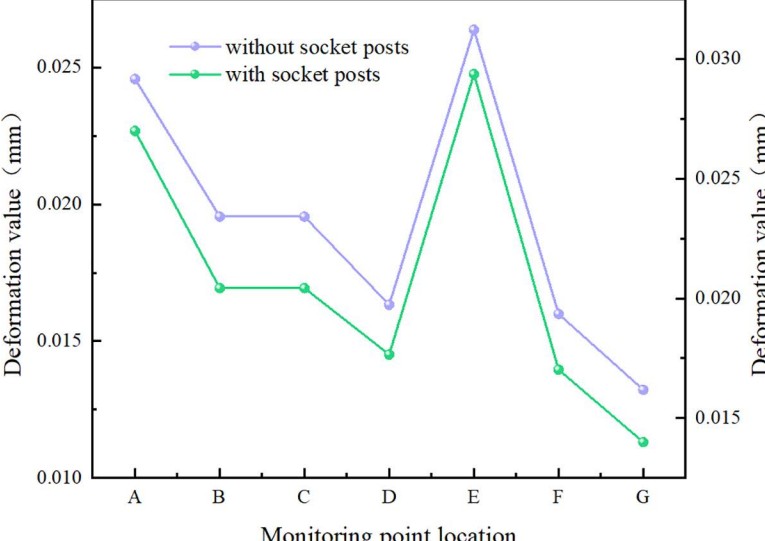

**Fig 24. Map comparing the position values of monitoring points.**

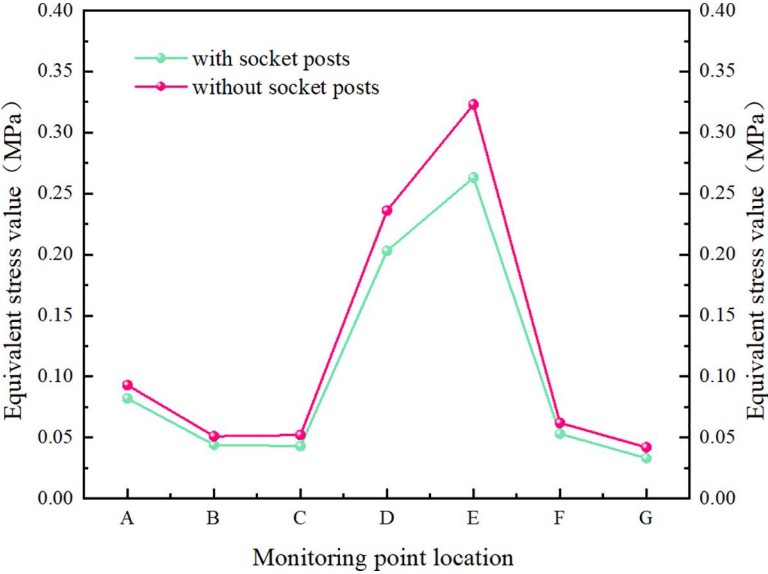

**Fig 25. Diagram comparing the equivalent stress values at monitoring points.**

D. The horizontal position of the joint between the side wall of the culvert body and the socket column.

E. Arc joint of the culvert body.

F. Central socket end.

G. Ends of both side socket columns.

H. The table below presents the stress results for two groups of distinct structures subjected to identical loading conditions.

The stress diagram results for the box culvert indicate that, under the same design load conditions and bearing capacity limit state, significant stress is observed at the middle connection end, the upper sides of both the left and right sections, and the side walls of the culvert body. The use of the socket column reduces the deformation and equivalent stress at the curved joint between the culvert body and the socket column. This demonstrates that the mortice and tenon structures at the culvert joints not only limit transverse and longitudinal displacement, but also prevent misalignment and uneven settlement. Additionally, they facilitate smooth load transfer, enhance the shear strength of the joint, and significantly improve the mechanical performance and practicality of the culvert body.

## 6. Conclusion

The comprehensive simulation and optimization results lead to the following conclusions regarding the Z-shaped assembled mortise and tenon splicing box culvert:

(1) The maximum deformation of the roof occurs at the center of the cross-section and gradually increases from the mid-span to the culvert's end section. The deformation of the middle section of the single culvert section is 75.86% higher than that of the end section. The maximum deformation and stress concentration in the side wall occur at the ends and the junctions of the culvert joints. The deformation at the maximum deformation of the side wall of the single culvert section is 980% higher than that at the end.

(2) Under identical filling conditions, the most significant difference in mechanical properties between the integral and assembled box culverts occurs at the joints of the prefabricated culvert, while the deformation and stress values at other locations are not significantly different. The deformation of the assembled lap culvert at the joint is 224.62% higher than that of the integral box culvert, while the equivalent stress is 93.51% higher; however, these values are relatively small.

(3) Considering that soil pressure increases with depth, the 1.8 m height and 0.5 m length of the joint significantly enhance the mechanical properties of the splicing culvert. The proposed design scheme should be implemented, provided that construction and prefabrication are conducted reasonably.

(4) Incorporating a 125 mm mortise and tenon joint structure into the prefabricated box culvert joint can restrict both horizontal and vertical displacements between the joints, enhance shear strength at the connections, and thereby improve the overall stability of the culvert.

## Author contributions

**Conceptualization:** Yuntao Yang, Longqi Liu.

**Data curation:** Yuntao Yang, Baolei Zhu, Bingjie Song.

**Formal analysis:** Yuntao Yang, Longqi Liu.

**Methodology:** Yuntao Yang.

**Project administration:** Longqi Liu, Hong Zhang, Baolei Zhu.

**Resources:** Longqi Liu, Baolei Zhu, Bingjie Song.

**Software:** Yuntao Yang, Baolei Zhu, Bingjie Song.

**Visualization:** Yuntao Yang.

**Writing – original draft:** Yuntao Yang, Bingjie Song.

**Writing – review & editing:** Longqi Liu, Hong Zhang.

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
