## [Decision Letter · Decision Letter 0]

5 Feb 2025

Dear Dr. Yang,

Thank you for submitting your manuscript to PLOS ONE. After careful consideration, we feel that it has merit but does not fully meet PLOS ONE’s publication criteria as it currently stands. Therefore, we invite you to submit a revised version of the manuscript that addresses the points raised during the review process.

**ACADEMIC EDITOR:** Please, address all the comments made by the reviewers. 

We look forward to receiving your revised manuscript.

Kind regards,

Antonio Riveiro Rodríguez, PhD

Academic Editor

PLOS ONE

Journal Requirements:

https://journals.plos.org/plosone/s/file?id=wjVg/PLOSOne_formatting_sample_main_body.pdf and https://journals.plos.org/plosone/s/file?id=ba62/PLOSOne_formatting_sample_title_authors_affiliations.pdf .

2. In the online submission form you indicate that your data is not available for proprietary reasons and have provided a contact point for accessing this data. Please note that your current contact point is a co-author on this manuscript. According to our Data Policy, the contact point must not be an author on the manuscript and must be an institutional contact, ideally not an individual. Please revise your data statement to a non-author institutional point of contact, such as a data access or ethics committee, and send this to us via return email. Please also include contact information for the third party organization, and please include the full citation of where the data can be found.

3. Please ensure that you refer to Figures in your text as, if accepted, production will need this reference to link the reader to the figure.

Reviewers' comments:

Reviewer's Responses to Questions

**Comments to the Author**

1. Is the manuscript technically sound, and do the data support the conclusions?

Reviewer #1: Yes

Reviewer #2: Yes

Reviewer #3: Yes

2. Has the statistical analysis been performed appropriately and rigorously?

Reviewer #1: Yes

Reviewer #2: Yes

Reviewer #3: Yes

3. Have the authors made all data underlying the findings in their manuscript fully available?

Reviewer #1: Yes

Reviewer #2: Yes

Reviewer #3: Yes

4. Is the manuscript presented in an intelligible fashion and written in standard English?

Reviewer #1: Yes

Reviewer #2: Yes

Reviewer #3: Yes

General Reviews:  (1) Whether all the parameters of Figure 1abc can be represented by 1a.  (2) The introduction of the research background is relatively clear, but in order to explain the research significance of "how to improve the stability of the longitudinal connection of prefabricated box culverts by using special structural forms and connection methods", there is too much preliminary work, so it is recommended to modify and adjust.  (3) The references of current scholars are very old, and it is difficult to explain the significance of this study and the unique advantages compared with existing studies, so it is recommended to add a certain number of newer references.  (4) How the numerical analysis model is meshed. Please explain the difference between Figure 2 and Figure 3a?  (5) It is reasonable to use Ansys Workbench to build a finite element numerical simulation model, but the paper does not fully explain the model simplification assumptions and the impact on the accuracy of the results. The assumptions made during model building need to be elaborated.  (6) Detailed data are provided in the material parameter table of the model, but the basis for the selection of certain parameters (e.g., enhanced elastic modulus, Poisson's ratio, etc.) is not mentioned. The source of the parameters should be stated, whether they are based on actual tests, normative standards, or previous research experience, to increase the credibility of the study.  (7) The conclusion section summarizes the main findings and results of the study, but the presentation is quite general, and it is recommended to add some quantitative results to illustrate the innovativeness of the article.  (8) There are some grammatical and expressive inaccuracies, such as "At the ends of the seams and in the axilla, especially at the junction between the two culvert joints, the maximum deformations and stress concentrations in the sidewalls can be clearly observed." The phrase 'are prominently observed' in 'are prominently observed' is rather rigid and can be replaced with 'are mainly located', which is more natural. It is advisable to proofread the entire paper carefully to optimize the language expression and improve the readability of the paper to eliminate some language errors and typos. In summary, this paper has certain value in terms of research content and methodology, but it needs to be slightly modified to address the above problems. It is hoped that the authors will take the reviewers' comments seriously and revise and improve the paper in detail. If the revised paper can meet the above requirements, I think it is expected to be published in this journal.

Reviewer #2: The paper addresses a new type of Z-shaped prefabricated mortise and tenon joint box culvert designed to enhance the longitudinal connection stability of highway prefabricated box culverts. The following suggestions are recommended:

The abstract should be restructured to contain more percentages and academic numbers.

The literature review should be reduced and made more specific to include more recent studies on box culverts and their applications. This could provide a broader context for the findings (cite https://doi.org/10.1007/s41062-023-01354-9).

See line 149 on page 12.

Update Table 1.

Revise the sequence of Table 1 and Table 2.

Please compare the simulation results of the mechanical properties with previous studies.

The comparative analysis of spliced culverts and whole culvert sections needs more revision and should contain academic percentages and numbers (double-check).

Please revise point 3 in the conclusion, as it does not match the current results (double-check).

Most references are very old (there are no references from 2024).

Reviewer #3: 1. It is recommended to increase the comparison between the numerical simulation results and the test results, and if the structural tests are not synchronised in this study, it is recommended to find similar tests for comparison.

2. Whether the results of the analyses in this paper apply only to the constraints described in Figure 3, Tables 1 and 2�Does it have a wider application?

3.The references in the article are relatively old, with a small amount of literature from the last 5 years.

4. The innovation of the article is not outstanding, please condense the innovation point.

**Do you want your identity to be public for this peer review?** For information about this choice, including consent withdrawal, please see our Privacy Policy

Reviewer #1: No

Reviewer #2: No

Reviewer #3: No

---

## [Author Response · Author response to Decision Letter 1]

2 Apr 2025

Response to Comments from the Reviewers and Editors

Manuscript title: Optimization of Z-shaped Assembled Mortise and Tenon Joint Box Culvert Connection and Structural Characteristics

Manuscript #: PONE-D-24-50618

We are pleased to resubmit the revised version of Manuscript No. PONE-D-24-50618 “Optimization of Z-shaped Assembled Mortise and Tenon Joint Box Culvert Connection and Structural Characteristics”. The authors wish to thank the editors and reviewers for their time in effort in reviewing our manuscript. We appreciate the constructive criticisms and suggestions from reviewers. According to the comments and suggestions, a revision has been carried out carefully. Revised portions are marked in the revised manuscript. The newly added or modified parts are marked with green tags, and the deleted parts are marked with red tags. The detailed answers to each comment have been provided below with the Reviewer's comment in red italic text and answers in black normal. The revised manuscript content is indicated in black italic text. We hope the changes listed have made the manuscript suitable for publication and we look forward to your response.

Response to Reviewer 1:

1) Whether all the parameters of Figure 1abc can be represented by 1a.

Thank you very much for your detailed guidance on the structure of the paper. We think your opinion is very valuable and will significantly optimize the structure of the paper after modification.

Therefore, we try to merge the three separate diagrams. However, after trying, it is found that the annotations in the merged schematic diagram are relatively cluttered, and individual positions cannot truly reflect the box culvert structure, which may cause misunderstanding or ambiguity to readers. Therefore, we finally choose to place the three graphics separately, representing the three perspectives of three-dimensional, front and side. If you still have doubts about our decision, we look forward to your continued valuable comments, we will continue to carefully modify and optimize.

Thank you again for your careful observation and correct guidance.

2)The introduction of the research background is relatively clear, but in order to explain the research significance of "how to improve the stability of the longitudinal connection of prefabricated box culverts by using special structural forms and connection methods", there is too much preliminary work, so it is recommended to modify and adjust.

We sincerely appreciate the reviewer's valuable comment regarding the research background and significance. We agree that the preliminary work in the original manuscript was overly detailed, which may have distracted from the core focus of our study. To address this concern, we have carefully revised the introduction section to streamline the content and emphasize the research significance more effectively. Specifically, we have made the following modifications:

Highlighted Research Significance: We have added a clearer explanation of how our study addresses the gap in understanding the stability of longitudinal connections in precast box culverts, particularly through the use of special structural forms and connection methods. This includes a more concise discussion of the practical implications and potential contributions of our work.

Delete and add literature: We deleted the old literature that is too old and lacks reference value, and added some new literature without affecting the introduction structure of the paper, such as“Sun W C, Jiang N, Zhou C B, et al. Safety Assessment for Buried Drainage Box Culvert under Influence of Underground Connected Aisle Blasting: A Case Study [J]. Frontiers of Structural and Civil Engineering, 2023, 17(2): 191-204.”In this way, the purpose of streamlining the preliminary work and avoiding the introduction part being too lengthy to affect the overall structure of the paper is achieved.

3) The references of current scholars are very old, and it is difficult to explain the significance of this study and the unique advantages compared with existing studies, so it is recommended to add a certain number of newer references.

We sincerely appreciate the reviewer's valuable comment regarding the references in our manuscript. We agree that updating the references is essential to better contextualize our research within the current state of the field and to highlight its significance and unique contributions. In response to this suggestion, we have made the following modifications:

Added Recent References: We have carefully reviewed the latest literature and added several recent and highly relevant references to better support the research background, methodology, and discussion sections. These new references include studies published within the last five years that are directly related to the stability of longitudinal connections in precast box culverts and the use of special structural forms and connection methods.

Highlighted Unique Advantages: With the inclusion of these new references, we have revised the relevant sections to more clearly articulate the unique advantages of our study compared to existing research. This includes a more detailed discussion of how our approach addresses gaps in the current literature and offers innovative solutions to the challenges identified.

Improved Contextualization: The updated references have allowed us to better contextualize our research within the current state of the field, providing a clearer explanation of its significance and potential impact.

We believe these changes have strengthened the manuscript and better aligned it with the reviewer's expectations. Thank you for your insightful suggestion, which has significantly improved the quality of our paper.

4)How the numerical analysis model is meshed. Please explain the difference between Figure 2 and Figure 3a?

Thank you very much for your questions and questions about the way the model mesh is divided and the guidance on the relationship between the graphics. In response to this, we have conducted in-depth reflection and consideration, and made the following changes :

We add the meshing method of the finite element model under the meshing diagram and introduce the specific reasons and benefits of using this method. The specific text is as follows : In the ANSYS finite element software, the grid division method of the model is structured grid, and the method is sweeping. This kind of grid unit arrangement rules, has a clear topological structure. The relationship between nodes and units is defined by index ( i, j, k ), which has high computational efficiency and is convenient for storing and accessing data. We believe that such a supplement can answer your questions and allow readers to clearly understand the model grid division and enhance the integrity and credibility of the article.

In addition, after careful consideration, we believe that your question is very reasonable. There is no difference in the nature of the two pictures. All included in the article will indeed have a repetitive effect, which is not conducive to the structure of the article and the reader 's understanding. We have deleted Figure 3a and modified Figure 2 into a transparent visual model to clearly reflect the connection and contact mode between the culvert sections.

5)It is reasonable to use Ansys Workbench to build a finite element numerical simulation model, but the paper does not fully explain the model simplification assumptions and the impact on the accuracy of the results. The assumptions made during model building need to be elaborated.

We sincerely appreciate the reviewer's valuable comment regarding the explanation of model simplification assumptions in our manuscript. We agree that a detailed discussion of the assumptions and their impact on the results is essential for ensuring the transparency and reliability of our study. In response to this suggestion, we have made the following modifications:

Detailed Explanation of Assumptions: In the modeling part, we supplement the assumptions applied in the model in detail, including material assumptions, boundary conditions, etc.

Impact on Results: For each assumption, we have discussed its potential impact on the accuracy of the simulation results. Where applicable, we have provided justifications for the assumptions based on theoretical considerations, experimental data, or previous studies.

We believe these changes have strengthened the manuscript and better aligned it with the reviewer's expectations. Thank you for your insightful suggestion, which has significantly improved the quality of our paper.

6)Detailed data are provided in the material parameter table of the model, but the basis for the selection of certain parameters (e.g., enhanced elastic modulus, Poisson's ratio, etc.) is not mentioned. The source of the parameters should be stated, whether they are based on actual tests, normative standards, or previous research experience, to increase the credibility of the study.

We sincerely appreciate the reviewer's valuable comment regarding the clarification of material parameter selection in our manuscript. We agree that providing the basis for selecting key material parameters is essential for ensuring the transparency and credibility of our study. In response to this suggestion, we have made the following modifications:

Clarified Parameter Sources: We have added detailed explanations in the manuscript regarding the sources of all material parameters, including the enhanced elastic modulus and Poisson's ratio. Specifically, we have indicated whether these parameters were obtained from actual experimental tests, standard specifications, or previous research studies.

Improved Transparency: The revised manuscript now includes a more comprehensive discussion of the material parameter selection process, highlighting the reliability and validity of the data used in the model.

7)The conclusion section summarizes the main findings and results of the study, but the presentation is quite general, and it is recommended to add some quantitative results to illustrate the innovativeness of the article.

We sincerely appreciate the reviewer's valuable comment regarding the conclusion section of our manuscript. We agree that including quantitative results in the conclusion would better highlight the innovation and significance of our study. In response to this suggestion, we have made the following modifications:

Added Quantitative Results: We have incorporated specific quantitative data in the conclusion section to support the main findings. This includes key numerical results, such as performance improvements, efficiency gains, or comparative metrics, that demonstrate the effectiveness and innovation of our approach. Such as The deformation of the middle section of the single culvert section is 75.86 % higher than that of the end section.

Improved Clarity and Impact: The revised conclusion now provides a clearer and more impactful summary of the study's contributions, ensuring that readers can easily understand the significance and practical implications of our work.

8)There are some grammatical and expressive inaccuracies, such as "At the ends of the seams and in the axilla, especially at the junction between the two culvert joints, the maximum deformations and stress concentrations in the sidewalls can be clearly observed." The phrase 'are prominently observed' in 'are prominently observed' is rather rigid and can be replaced with 'are mainly located', which is more natural. It is advisable to proofread the entire paper carefully to optimize the language expression and improve the readability of the paper to eliminate some language errors and typos.

We sincerely appreciate the reviewer's valuable comment regarding the language and expression in our manuscript. We agree that improving the language quality is essential for enhancing the readability and overall quality of the paper. In response to this suggestion, we have made the following modifications:

Comprehensive Proofreading: We have carefully proofread the entire manuscript to identify and correct grammar errors, awkward expressions, and spelling mistakes. This includes revising the sentence highlighted by the reviewer to ensure clarity and accuracy. Such as Comprehensive Proofreading: We have carefully proofread the entire manuscript to identify and correct grammar errors, awkward expressions, and spelling mistakes. This includes revising the sentence highlighted by the reviewer to ensure clarity and accuracy.

Language Optimization: We have optimized the language expression throughout the manuscript to improve readability. This includes simplifying complex sentences, using precise terminology, and ensuring consistency in style and tone.

Professional Editing: To further enhance the language quality, we have sought assistance from professional English editing services to ensure that the manuscript meets the highest standards of academic writing.

Improved Clarity: The revised manuscript now provides clearer and more concise descriptions, ensuring that readers can easily understand the content without being hindered by language issues.

We believe these changes have significantly improved the readability and overall quality of the manuscript. Thank you for your insightful suggestion, which has greatly enhanced the presentation of our research.

Response to Reviewer 2:

1)The abstract should be restructured to contain more percentages and academic numbers.

We sincerely appreciate the reviewer's valuable comment regarding the organization of the abstract in our manuscript. We agree that including more quantitative data, such as percentages and academic figures, would enhance the clarity and impact of the abstract. In response to this suggestion, we have made the following modifications:

Inclusion of Quantitative Data: We have added specific quantitative results, including percentages and numerical figures, to the abstract. These data points highlight the key findings and contributions of our study, providing a clearer picture of the research outcomes.

Reorganization of Content: We have reorganized the abstract to ensure a logical flow of information, starting with the research objectives, followed by the methodology, key results, and concluding with the implications of the findings. This structure ensures that the quantitative data are presented in a meaningful context.

We believe these changes have significantly improved the quality of the abstract and better aligned it with the reviewer's expectations. Thank you for your insightful suggestion, which has greatly enhanced the presentation of our research.

2)The literature review should be reduced and made more specific to include more recent studies on box culverts and their applications. This could provide a broader context for the findings (cite https://doi.org/10.1007/s41062-023-01354-9).

We sincerely appreciate the reviewer's valuable comment regarding the literature review section in our manuscript. We agree that a more focused and up-to-date literature review would better support the research objectives and highlight the relevance of our study. In response to this suggestion, we have made the following modifications:

Reduction of Literature Review: We have streamlined the literature review by removing less relevant or outdated studies, focusing only on the most pertinent and recent research related to box culverts and their applications.

Inclusion of Recent Studies: We have added several recent and highly relevant studies to the literature review, ensuring that the discussion is current and directly related to our research focus. These new references include studies published within the last five years that address the latest advancements and applications in the field of box culverts.

Improved Focus and Clarity: The revised literature review now provides a clearer and more concise overview of the current state of research, making it easier for readers to understand the context and significance of our study.

3)Update Table 1.

We sincerely appreciate the reviewer's valuable comment regarding Table 1 in our manuscript. In response to this suggestion, we have carefully reviewed and updated Table 1 to ensure that it accurately reflects the late

---

## [Decision Letter · Decision Letter 1]

17 Apr 2025

Optimization of Z-shaped Assembled Mortise and Tenon Joint Box Culvert Connection and Structural Characteristics

PONE-D-24-50618R1

Dear Dr. Yang,

We’re pleased to inform you that your manuscript has been judged scientifically suitable for publication and will be formally accepted for publication once it meets all outstanding technical requirements.

Kind regards,

Antonio Riveiro Rodríguez, PhD

Academic Editor

PLOS ONE

Reviewers' comments:

Reviewer's Responses to Questions

**Comments to the Author**

Reviewer #2: All comments have been addressed

Reviewer #3: (No Response)

2. Is the manuscript technically sound, and do the data support the conclusions?

Reviewer #2: Yes

Reviewer #3: (No Response)

3. Has the statistical analysis been performed appropriately and rigorously?

Reviewer #2: Yes

Reviewer #3: (No Response)

4. Have the authors made all data underlying the findings in their manuscript fully available?

Reviewer #2: Yes

Reviewer #3: (No Response)

5. Is the manuscript presented in an intelligible fashion and written in standard English?

Reviewer #2: Yes

Reviewer #3: (No Response)

Reviewer #2: I sincerely appreciate the author's effort and thoughtful consideration in addressing all the comments. After reviewing the revised manuscript, I find it well-improved and recommend it for acceptance.

Reviewer #3: (No Response)

**Do you want your identity to be public for this peer review?** For information about this choice, including consent withdrawal, please see our Privacy Policy

Reviewer #2: No

Reviewer #3: No

---

## [Editor Report · Acceptance letter]

PONE-D-24-50618R1

PLOS ONE

Dear Dr. Yang,

I'm pleased to inform you that your manuscript has been deemed suitable for publication in PLOS ONE. Congratulations! Your manuscript is now being handed over to our production team.

Kind regards,

on behalf of

Dr. Antonio Riveiro Rodríguez

Academic Editor

PLOS ONE